# On periodic solutions associated with the nonlinear feedback loop in the non-dissipative Lorenz model

B.-W. Shen

Department of Mathematics and Statistics, San Diego State University, 5500 Campanile Drive, San Diego, CA, 92182, USA

Correspondence to: B.-W. Shen (bshen@mail.sdsu.edu; bowen.shen@gmail.com)

Abstract. In this study, we discuss the role of the linear heating term and nonlinear terms associated with a nonlinear feedback loop in the energy cycle of the three-dimensional (X-Y-Z) nondissipative Lorenz model (3D-NLM), where (X, Y, Z) represent the solutions in the phase space. Using trigonometric functions, we first present the closed-form solution of the nonlinear equation

- $d^2X/d\tau^2 + (X^2/2)X = 0$  without the heating term (i.e., rX), (where  $\tau$  is a non-dimensional time and r is the normalized Rayleigh number), a solution that has not been previously documented. Since the solution of the simplfied 3D-NLM is oscillatory (wave-like) and since the nonlinear term ( $X^3$ ) is associated with the nonlinear feedback loop, here, we suggest that the nonlinear feedback loop may act as a restoring force. When the heating term is considered, the system yields three critical
- points. A linear analysis suggests that the origin (i.e., the trivial critical point) is a saddle point and that the other two non-trivial critical points are stable. Here, we provide an analysis for three types of solutions that are associated with these critical points. Two of the solutions represent closed curves that travel around one non-trivial critical point or all three critical points. The third type of solution, appearing to connect the stable and unstable manifolds of the saddle point, is called the
- homoclinic orbit. Using the solution that contains one non-trivial critical point, here, we show that the competing impact of the nonlinear restoring force and the linear (heating) force determines the partitions of the averaged available potential energy from the Y and Z modes. Based on the energy analysis, an energy cycle with four different regimes is identified. The cycle is only half of a "large" cycle, displaying the wing pattern of a glasswinged butterfly. The other half cycle is antisymmetric
- with respect to the origin. The two types of oscillatory solutions with either a small cycle or a large cycle are orbitally stable. As compared to the oscillatory solutions, the homoclinic orbit is not periodic because it "takes forever" to reach the origin. Two trajectories with starting points near the homoclinic orbit may be diverged because one moves with a small cycle and the other moves with

a large cycle. Therefore, the homoclinic orbit is not orbitally stable. In a future study, dissipation
and/or additional nonlinear terms will be included in order to determine how their interactions with
the original nonlinear feedback loop and the heating term may change the periodic orbits (as well as
homoclinic orbits) to become quasi-periodic orbits and chaotic solutions.

#### 1 Introduction

It has been fifty years since Lorenz published his breakthrough modeling study (Lorenz, 1963) 30 that changed our views on the predictability of weather and climate (Solomon et al., 2007). The model has been extensively investigated by researchers in various fields including earth science, engineering, mathematics, philosophy, and physics (e.g., Gleick, 1987; Sprott, 2003; Jordan and Smith, 2007; Anthes, 2011; Hirsch et al., 2013; Strogatz , 2015). Lorenz's model with three Fourier modes, which represents the solution to the 2-D Rayleigh–Benard equation (Saltzman,

1962; Lorenz, 1963), is known as the three-dimensional Lorenz model (3DLM). In this paper, we use 3D-NLM to refer to the non-dissipative version of the model that is introduced later in the text.

The scientific community agrees that weather is chaotic, with a finite predictability, and that the source of chaos is nonlinearity. Since the degree of nonlinearity is finite within the 3DLM, the impact of increased nonlinearity on system solutions and/or their stability has been studied

- using generalized LMs with additional Fourier modes (e.g., Curry, 1978; Curry et al., 1984; Howard and Krishnamurti, 1986; Hermiz et al., 1995; Thiffeault and Horton, 1996; Musielak et al., 2005; Roy and Musielak, 2007a). As compared to the 3DLM, some of the generalized LMs have suggested larger Rayleigh number values (or heating parameters) for the onset of chaos, while others have indicated smaller values. This discrepancy may be attributed to different mode truncations (e.g.,
- Curry et al., 1984; Thiffeault and Horton, 1996; Roy and Musielak, 2007a, b, c; Shen, 2014, 2015, 2016) that lead to different degrees of nonlinearity and different systems whose energy may or may not be conserved (e.g., Roy and Musielak, 2007a).

Among studies using generalized LMs, the pioneering study of Prof. Curry (Curry et al., 1984) suggested that chaotic responses disappeared when sufficient modes are included. Recent studies

- by Prof. Musielak and his colleagues (Musielak et al., 2005; Roy and Musielak, 2007a, b, c) have examined the transition to chaos and the fractal dimensions of generalized LMs, and have emphasized the importance of proper mode truncation in energy conservation. More recent studies (Shen, 2014, 2015, 2016) have discussed the importance of proper Fourier mode selection in extending the nonlinear feedback loop of the 3DLM. The feedback loop, which consists of the
- two nonlinear terms of the 3DLM, includes a pair of downscale and upscale transfer processes associated with the Jacobian function of the partial differential equation, discussed in Sect. 2. As previously suggested, the original feedback loop may help stabilize the solution for 1

- negative nonlinear feedback that produces non-trivial stable critical points when r < 42.9 within a</li>
  five-dimensional LM and when r < 116.9 within a seven-dimensional LM. Based on the above, it has been hypothesized that a system's stability can be improved further with additional modes that can provide negative nonlinear feedback. While the importance of an increased degree of nonlinearity with more Fourier modes has been discussed in recent studies, the competing role of the nonlinear terms with the linear (heating) term and/or dissipative terms deserves to be examined</li>
- in order to ascertain the conditions under which nonlinear processes may lead to stable or chaotic solutions.

Roupas (2012) and others have indicated that the 3DLM in the dissipative limit, which is referred to as the 3D-NLM, contains two conserved quantities that represent the conservation of ( $\overline{\text{KE}} + \overline{\text{PE}}$ ) and ( $\overline{\text{KE}} + \overline{\text{APE}}$ ), respectively. Here,  $\overline{\text{KE}}$ ,  $\overline{\text{PE}}$ , and  $\overline{\text{APE}}$  are the domain-averaged kinetic energy,

- potential energy, and available potential energy, respectively. These two quantities,  $(\overline{\text{KE}} + \overline{\text{PE}})$  and  $(\overline{\text{KE}} + \overline{\text{APE}})$ , are related to the Nambu Hamiltonians (Nambu, 1973; Nevir and Blender, 1994; Floratos, 2011; Roupas, 2012; Blender and Lucarini, 2013). As a result of conservation properties, the collective impact of the nonlinear feedback loop and the linear (heating) term may effectively act as a "restoring" force. The simplicity of the 3D-NLM enables an examination of how the nonlinear
- feedback loop and the linear (heating) term work together to produce oscillatory solutions (in the phase space). In this work, we address this issue in conjunction with how the available potential energy is partitioned amongst two different Fourier modes, Y and Z, where Z is included in order to enable the nonlinear feedback loop.

The paper is organized as follows: In Sect. 2, we present the governing equations and the 3D-

- NLM, introduce the nonlinear feedback loop, and derive the energy conservation laws. In Sect. 3, we illustrate the role of the nonlinear feedback loop (with r = 0) in acting as a restoring force. With inclusion of the heating term, as well as the nonlinear feedback loop, we present a linear stability analysis for the three critical points and then discuss three types of solutions using analytical and numerical methods. The solutions include two types of oscillatory solutions and the so-called
- homoclinic orbit (Jordan and Smith, 2007; Sprott, 2003). An energy cycle with four regimes is analyzed based on the tendency of  $\overline{\text{KE}}$  and the partition of  $\overline{\text{APE}}$  at different scales (i.e., either Y or Z). Concluding remarks are provided at the end. Appendix A discusses the derivation of the 3D-NLM, and Appendix B presents a closed-form solution to the system with r = 0 using elliptic functions (e.g., Davis, 1960). Appendix C discusses the equations and initial conditions that are used
- to obtain the solution of the homeclinic orbit.

9

### 2 Governing equations and the non-dissipative Lorenz model

The following governing equations for a 2D (x, z), Boussinesq flow are introduced in order to derive the non-dissipative Lorenz model (3D-NLM) and to calculate its kinetic and potential energy:

$$\frac{\partial}{\partial t}\nabla^2 \psi = -J(\psi, \nabla^2 \psi) + g\alpha \frac{\partial \theta}{\partial x} + \nu \nabla^4 \overline{\psi}, \tag{1}$$

5 
$$\frac{\partial\theta}{\partial t} = -J(\psi,\theta) + \frac{\Delta T}{H} \frac{\partial\psi}{\partial x} + \kappa \nabla^2 \theta,$$
 (2)

where  $\psi$  is the streamfunction that yields  $u = -\psi_z$  and  $w = \psi_x$  that represent the horizontal and vertical velocity perturbations, respectively, and  $\theta$  is the temperature perturbation.  $\Delta T$  represents the temperature difference between the bottom and top boundaries. The constants, g,  $\alpha$ ,  $\nu$ , and  $\kappa$ 

denote the acceleration of gravity, the coefficient of thermal expansion, the kinematic viscosity, and the thermal diffusivity, respectively. The Jacobian of two arbitrary functions is defined as  $J(A, B) = (\partial A/\partial x)(\partial B/\partial z) - (\partial A/\partial z)(\partial B/\partial x)$ . The crossout symbol indicates the negligence of a term in the dissipationless limit. Equations (1) and (2), with the dissipative terms, were first used in Saltzman (1962) and Lorenz (1963).

The non-dissipative Lorenz model (3D-NLM) is written as:

$$\frac{\mathrm{d}X}{\mathrm{d}\tau} = \sigma Y,\tag{3}$$

$$\frac{\mathrm{d}Y}{\mathrm{d}\tau} = -XZ + rX,\tag{4}$$

$$\frac{\mathrm{d}Z}{\mathrm{d}\tau} = XY.\tag{5}$$

- Here, (X,Y,Z) represent the amplitude of the Fourier modes. τ = κ(1 + a<sup>2</sup>)(π/H)<sup>2</sup>t is the dimensionless time. a is the ratio of the vertical scale of the convection cell to its horizontal scale. H is the domain height, and 2H/a represents the domain width. σ = ν/κ is the Prandtl number, and r = R<sub>a</sub>/R<sub>c</sub> is the normalized Rayleigh number or the heating parameter. R<sub>a</sub> is the Rayleigh number, R<sub>a</sub> = gαH<sup>3</sup>ΔT/νκ, and R<sub>c</sub> is the critical value for the free-slip Rayleigh–Benard problem,
  R<sub>c</sub> = π<sup>4</sup>(1 + a<sup>2</sup>)<sup>3</sup>/a<sup>2</sup>. The "forcing" terms on the right-hand side of Eqs. (4) and (5) are referred to
- as the linear force, or the heating terms on the right-hand side of Eqs. (4) and (5) are referred to as the linear force, or the heating term (rX), and the nonlinear force terms (-XZ and XY). Note that as a result of scale selection in the original Lorenz model, the appearance of  $\sigma Y$  comes from  $g\alpha \frac{\partial \theta}{\partial x}$ , which is not associated with the dissipative terms. Appendix A provides detailed derivations. The 3D-NLM is integrated forward in time using the fourth-order Runge–Kutta scheme. Without
- a loss of generality, only three different values of the normalized Rayleigh number, r (r = 0, r = 25, and r = 45) are used, while keeping other parameters, including  $\sigma = 10$  and  $a = 1/\sqrt{2}$ , constant. A dimensionless time interval ( $\Delta \tau$ ) of 0.01 is used and the total number of time steps is 10 000, giving a total dimensionless time ( $\tau$ ) of 100. A smaller  $\Delta \tau$  is used to improve accuracy. In this study, unless stated otherwise, the initial conditions (ICs) are as follows:

$$(X,Y,Z) = (0,1,0).$$

(6)

These settings were used in order to examine the stability of the 5DLM and 6DLM in Shen (2014, 2015), who also discussed the dependence of the solution on different r and σ. To illustrate the unique features of solutions, a very small time step and/or different ICs are used, including (X,Y,Z) = (±ε,0,0), (0,±ε,,0), and (2√σr,0,2r). The first IC is used for showing solutions with a small cycle, and the second is used for discussing solutions with a big cycle. The third IC is used to perform a numerical simulation for the homoclinic orbit (Jordan and Smith, 2007; Sprott, 2003).

# 2.1 The nonlinear feedback loop and energy conservation laws

Nonlinear terms in the 3D-NLM (and 3DLM) have been shown to result from the Jacobian term, J(ψ,θ), in Eq. (2). The nonlinear interaction of two wave modes via the Jacobian term can generate
or impact a third wave mode through a downscale (or upscale) transfer process. The subsequent

upscale (or downscale) transfer process associated with the third wave mode can provide feedback to the incipient wave mode(s). As illustrated in Shen (2014), XY and -XZ, respectively, represent the downscale and upscale transfer processes that form a nonlinear feedback loop. When new modes are properly included, the feedback loop can be extended. In the following subsections, we discuss
the role of the 3D-NLM nonlinear feedback loop in the energy conservation and partition of available

potential energy, which, in turn, helps produce oscillatory solutions.

The domain-averaged kinetic energy ( $\overline{\text{KE}}$ ), the potential energy ( $\overline{\text{PE}}$ ), and the available potential energy ( $\overline{\text{APE}}$ ) are defined as follows (e.g., Thiffeault and Horton, 1996; Blender and Lucarini, 2013; Shen, 2014, 2015):

$$\overline{\text{KE}} = \frac{1}{2} \int_{0}^{2H/a} \int_{0}^{H} (u^2 + w^2) \mathrm{d}z \mathrm{d}x,$$
 (7)

$$\overline{\text{PE}} = -\int_{0}^{2H/a} \int_{0}^{H} g\alpha(z\theta) dz dx,$$
(8)

$$\overline{\text{APE}} = -\frac{g\alpha H}{2\Delta T} \int_{0}^{2H/a} \int_{0}^{H} (\theta)^2 \mathrm{d}z \mathrm{d}x.$$
(9)

Through straightforward derivations, we obtain the following:

$$\overline{\mathrm{KE}} = \frac{C_o}{2} X^2, \tag{10}$$

$$\overline{\text{PE}} = -C_o \sigma Z,\tag{11}$$

$$\overline{\text{APE}} = -\frac{C_o}{2} \frac{\sigma}{r} (Y^2 + Z^2), \tag{12}$$

where  $C_o = \pi^2 \kappa^2 \left(\frac{1+a^2}{a}\right)^3$ .  $\overline{\text{APE}}$  is non-positive (i.e.,  $\overline{\text{APE}} \le 0$ ), since any perturbation reduces the 155 energy transformable to  $\overline{\text{KE}}$ .

1

Equations (10) and (11) lead to:

$$\overline{\mathrm{KE}} + \overline{\mathrm{PE}} = C_o \left(\frac{X^2}{2} - \sigma Z\right) = C_1, \tag{13}$$

while Eqs. (10) and (12) yield:

60 
$$\overline{\text{KE}} + \overline{\text{APE}} = \frac{C_o}{2} \left( X^2 - \frac{\sigma}{r} (Y^2 + Z^2) \right) = C_2.$$
 (14)

With Eqs. (3)–(5), the time derivative for both Eqs. (13) and (14) is zero, so these two equations describe energy conservation. Both  $C_1$  and  $C_2$  are constants and are determined by the initial conditions. Thus, if we express Z and Y<sup>2</sup> as functions of X, they are single valued. To facilitate our

- discussions, the contribution to APE from an individual mode is defined as  $\overline{APE_I} = -C_o \sigma I^2/2r$ , where I = Y or Z; therefore,  $\overline{APE} = \overline{APE_Y} + \overline{APE_Z}$ . Note that Eqs. (13) and (14) are related to the two Nambu Hamiltonians,  $C = -X^2/2 + \sigma Z$  and  $H = Y^2/2 + Z^2/2 - rZ$  (Nambu, 1973; Nevir and Blender, 1994; Floratos, 2011; Roupas, 2012; Blender and Lucarini, 2013).
- From the initial conditions in Eq. (6), we have  $C_1 = 0$  and  $C_2/C_o = -\sigma/2r$ , the latter of which 170 is -0.2 for r = 25 and -0.11 for r = 45. Figure 1 provides the time evolution of the conserved quantities:  $(\overline{\text{KE}} + \overline{\text{PE}})$  and  $(\overline{\text{KE}} + \overline{\text{APE}})$  in Eqs. (13) and (14) from the 3D-NLM. At a larger r (e.g., r = 45), a finer  $\Delta \tau$  is required to improve the numerical accuracy of simulated total energy (Fig. 1c). In this study, to facilitate discussions and unless stated otherwise, either  $C_1$  or  $C_2$  is assumed to be zero.

## 175 3 Discussions

In this section, we discuss the competing role of the nonlinear terms and the linear forcing term in the transient solutions and the energy cycle of the 3D-NLM. From Eqs. (3), (4), and (13), we obtain

$$\frac{\mathrm{d}^2 X}{\mathrm{d}\tau^2} + M^2 X = 0,\tag{15a}$$

and

$$M^{2} = \frac{X^{2}}{2} - \left(\sigma r + \frac{C_{1}}{C_{o}}\right).$$
(15b)

The three terms on the right-hand side of Eq. (15b) represent the impact of nonlinearity, the linear (heating) force, and the initial conditions, respectively. Their competing impacts (i.e., their differences) determine the sign of M<sup>2</sup>, and, thus, the characteristics of the solution. Based on the relative magnitude of the initial state that may lead to (σr + C<sub>1</sub>/C<sub>o</sub>) ≤ 0 or > 0, two types of solutions can be identified (Roupas, 2012). In this study, we focus on the case with 0 ≤ (σr+C<sub>1</sub>/C<sub>o</sub>) by using C<sub>1</sub> = 0 in Sect 3.2 and C<sub>1</sub>/C<sub>o</sub> = X<sub>o</sub><sup>2</sup>/2 in Sect 3.3, where X<sub>o</sub> is the initial value of X. To understand the role of the nonlinear terms (i.e., the nonlinear feedback loop), we begin discussions

<sup>190</sup> by solving the solution to the equation that contains no nonlinear terms.

Nonlinear Processes in Geophysics Discussions

## 3.1 Solutions of the linear system without the nonlinear feedback loop

By assuming no nonlinear terms, we can begin with two equations:  $dX/d\tau = \sigma Y$  and  $dY/d\tau = rX$ and only one conservation law, as follows:

$$\overline{\mathrm{KE}} + \overline{\mathrm{APE}} = \frac{C_o}{2} \left( X^2 - \frac{\sigma}{r} Y^2 \right) = C_2.$$
(16)

This linear case yields  $M^2 = -\sigma r$ , and Eq. (15) becomes  $d^2 X/d\tau^2 - \sigma r X = 0$ . A local analysis suggests that the origin, the only critical point in the linear system, is a saddle point. After straightforward derivations, the solution is:

$$X = X_1 e^{-\sqrt{\sigma r \tau}} + X_2 e^{+\sqrt{\sigma r \tau}},\tag{17}$$

- where  $X_1$  and  $X_2$  are constant coefficients. The origin (X,Y) = (0,0) is a saddle point, and the trajectory is hyperbolic with solutions exhibiting exponential growth and decay. The initial condition, which determines  $dX/dY (= \sigma Y/rX)$ , can help select the proper mode(s). For example,  $(X,Y) = (\sqrt{\sigma/r}, 1)$  only provides the growing mode with  $(X_1, X_2) = (0, \sqrt{\sigma/r})$ , while (X,Y) = $(\sqrt{\sigma/r}, -1)$  leads to the decaying mode with  $(X_1, X_2) = (\sqrt{\sigma/r}, 0)$ . The former and latter display
- the properties of unstable and stable manifolds, respectively (Ide et al., 2002). For the nonlinear case, a "current" state (i.e., ICs) may vary with time. Therefore, either mode may appear at different stages. For example, as shown with the homoclinic orbit in Sect. 3.3, a trajectory with an initial point of  $(X, Y, Z) = (2\sqrt{\sigma r}, 0, 2r)$  may approach the origin at a decay rate of  $\sqrt{\sigma r}$ , while a trajectory beginning near the origin may depart at a growth rate of  $\sqrt{\sigma r}$ . Based on Eqs. (16) and (17), although
- the time change of  $(\overline{\text{KE}} + \overline{\text{APE}})$  remains zero, the  $\overline{\text{KE}}$  produced using only the linear (heating) force has no upper limit. Such an outcome could violate the linear assumption, and, thus, nonlinearity should be included.

#### **3.2** Nonlinear solutions and the nonlinear restoring force for r = 0

Here, we examine the impact of nonlinear terms using a special case with r = 0 and (X, Y, Z) =215 (0,1,0), leading to  $M^2 = X^2/2$ . Thus, Eq. (15) becomes:

$$\frac{\mathrm{d}^2 X}{\mathrm{d}\tau^2} + \frac{X^2}{2} X = 0.$$
(18)

As compared to the case with  $r \neq 0$  in Eqs. (13) and (14), the energy conservation laws obtained using r = 0 are:

$$\frac{X^2}{2} - \sigma Z = 0,$$
 (19)

$$Y^2 + Z^2 = 1, (20)$$

which, in turn, lead to:

$$Y^2 + \frac{X^4}{4\sigma^2} = 1.$$
 (21)

Solutions to the above equation can be obtained as follows:

$$X^2 = 2\sigma \sin(\phi), \tag{22a}$$

$$Y = \cos(\phi), \tag{22b}$$

$$Z = \sin(\phi), \tag{22c}$$

where the phase function,  $\phi$ , can be determined from Eqs. (18) and (22a) and is written as:

$$\phi = \int_{0}^{\tau} X \,\mathrm{d}\tau_o, \tag{23a}$$

which can also be displayed as:

\_

$$235 \quad \phi = \int_{0}^{\tau} \int_{0}^{\tau} \sigma Y \mathrm{d}\tau_1 \mathrm{d}\tau_2. \tag{23b}$$

To illustrate the solution's characteristics, Eqs. (22b) and (23b) are solved using the following iterative method:

$$Y_n = \cos(\phi_n), \quad n = 0, 1, 2...N$$
 (24a)

$$\phi_{n+1} = \int_{0}^{\tau} \int_{0}^{\tau} \sigma Y_n \mathrm{d}\tau_1 \mathrm{d}\tau_2,$$
 (24b)

where N is the number of iterations. Over a period of time, an initial guess for the phase function is given as  $\phi_0(\tau) = \tau$ . We insert the first phase function,  $\phi_0$ , into Eq. (24a) to obtain  $Y_0$ . We then calculate the next phase function,  $\phi_1$ , using  $Y_0$  and Eq. (24b). The integral in Eq. (24b) is calculated

using the trapezoidal rule. We then repeat the above calculations N times. Numerical results using N = 100 are provided in Fig. 2. The phase function (Fig. 2a) oscillates with time and varies between 0 and π, consistent with Eq. (22a) as a result of sin(φ) ≥ 0. Therefore, the solution to Eq. (18) is oscillatory instead of growing or decaying exponentially (as shown in Figs. 2b and c). The nonlinear term in Eq. (18) may be viewed as a nonlinear restoring force. Such a suggestion is consistent with
the view (Shen, 2014) that the pair of nonlinear terms (-XZ and XY), leading to the nonlinear term (X<sup>2</sup>/2), can form a nonlinear feedback loop within the 3DLM. When σ = 10 is replaced by σ = 20, an oscillatory solution with a different period is obtained, as shown in Fig. 2d. As a result of the simple method for integral calculation, a fine Δτ may be required in order to obtain accurate solutions, as indicated by the red and green lines for the results obtained using Δτ = 0.0001 and

255 0.01, respectively (see Fig. 2a).

To verify the integral form of the solutions in Eqs. (24a) and (24b), the numerical solutions of the 3D-NLM using r = 0 (e.g., Eqs. 3–5) are provided in Fig. 3. In panel (a), the blue "dot" indicates the initial temporal evolution of the phase function that is calculated by performing a time integration of X using Eq. (23a), where X is obtained from the 3D-NLM; and the red line indicates

the phase function calculated using Eqs. (24a) and (24b). Both are in good agreement. The simulated trajectories in the X-Y and X-Z sections are elliptic and parabolic (Figs. 3b and c), respectively, consistent with the analytical relationships in Eqs. (21) and (19), respectively. Figure 3d provides the time evolution of oscillatory Y (red) and X (blue), consistent with the results provided in Figs. 2b and c, respectively. As indicated in Appendix B, the oscillatory characteristics of the solution can also be illustrated using elliptic functions.

## **3.3** Closed form solutions near a non-trivial critical point for $r \neq 0$

In the previous sections, we used Eq. (15) to illustrate the individual impact of the linear (heating) force and the nonlinear feedback loop using cases with  $M^2 < 0$  and  $M^2 \ge 0$ , respectively, neither of which changes the sign during the integration. Here, using the initial condition  $(X, Y, Z) = (X_o, 0, 0)$ 

- and  $0 < X_o$ , we consider a more general case with  $M^2 = (X^2/2 \sigma r X_o^2/2)$ , whose sign may vary during the time integration depending on the relative magnitude of the nonlinearity and the linear (heating) force.  $M^2$  has two zeros at  $X = \pm X_t$ , where  $X_t$  is defined as  $\sqrt{2\sigma r + X_o^2}$ . These are called turning points. Based on an analysis using the WKB approximation (Bender and Orszag, 1978), there appears to be a growing or decaying solution for  $|X| < X_t$  and an oscillatory (wave-
- like) solution for  $|X| > X_t$ . The former is largely impacted by the linear (heating) force while the latter is impacted by the nonlinear restoring force. Additional analyses are provided below.

To understand the stability of solutions, below, we present a local analysis by linearizing the system with respect to a non-trivial critical point. Using Eqs. (3-5), we first solve for the non-trivial critical point which is  $(X_c, Y_c, Z_c) = (X_c, 0, r)$ . Here,  $X_c$  can be any constant and is selected as the value of the turning point using the justification presented below. From Eqs. (3) and (15), the

280 the value of the turning point using the justification presented below. From Eqs. (3) and (15), the 3D-NLM system can be written as:

$$dX/d\tau = \sigma Y,\tag{25}$$

$$dY/d\tau = -\frac{M^2}{\sigma}X.$$
 (26)

The two equations, with an initial condition of  $(X, Y) = (X_0, 0)$ , lead to three critical points,  $(X_c, Y_c)$  at (0,0) and  $(\pm \sqrt{2\sigma r + X_0^2}, 0)$ . Note that the non-trivial critical points  $\pm X_c$  are turning points due to M = 0 in Eq. 15b.

Next, each of the total fields is decomposed into its basic part and perturbation, as follows:

$$V = V_c + V', \tag{27}$$

where V indicates the total fields (X, Y, Z),  $V_c$  represents the basic state obtained from the solution of the critical point, and V' is a perturbation that measures the departure from the critical point. Using the above equation, the linearized system corresponding to the nonlinear system from Eqs. 3-5 is:

$$\frac{ds}{dt} = As,\tag{28}$$

where s = (X', Y', Z') and the matrix A that is evaluated at the non-trivial critical point is written as:

$$A = \begin{pmatrix} 0 & \sigma & 0 \\ 0 & 0 & -X_c \\ 0 & X_c & 0 \end{pmatrix},$$
 (29)

The above system with Eqs. (28-29) yields eigenvalues of 0 and  $\pm iX_c$ , suggesting that the non-trivial critical point  $(X_t, 0, r)$  for the linearized system is a stable node as a center. The other non-trivial critical point  $(-X_t, 0, r)$  shares the same features. One important feature is the dependence of the

critical point  $(-X_t, 0, r)$  shares the same reatures. One important reature is the dependence of the solution's period  $(2\pi/X_c)$  on the initial condition  $(X_o)$ . Next, to illustrate its periodicity, we present a closed form solution for the nonlinear system.

Equations (13-14), with the initial conditions  $(X, Y, Z) = (X_o, 0, 0)$ , lead to the following two 305 equations:

$$\frac{X^2}{2} - \sigma Z = \frac{X_o^2}{2},$$
(30)

$$rX^2 - \sigma(Y^2 + Z^2) = rX_o^2.$$
 (31)

Using the same procedures as those provided in Sect. 3.2 (for r = 0), the following closed-form 310 solutions (for  $r \neq 0$ ) are obtained:

$$Y = rsin(\phi), \tag{32a}$$

$$Z = -r\cos(\phi) + r, \tag{32b}$$

$$X^2 = 2\sigma(r - r\cos(\phi)) + X_o^2, \qquad (32c)$$

330

$$\phi = \int X d\tau. \tag{32d}$$

Equations (32c) and (32d) are iteratively computed in order to determine the value of  $\phi$  that is used to calculate the solutions of X, Y, and Z. Note that because  $X_t^2 = 2\sigma r + X_o^2$ , Eqs. (32a) and (32c) lead to:

$$\left(\frac{X^2 - X_t^2}{2\sigma}\right)^2 + Y^2 = r^2.$$
(33)

Since  $r \ge rcos(\phi)$ ,  $X^2$  is always positive in Eq. (32c). Without a loss of generality,  $X = +\sqrt{2\sigma(r - rcos(\phi)) + X_o^2}$  is first discussed, yielding a minimum of  $X_{min} = X_o$  and a maximum of  $X_{max} = \sqrt{4\sigma r + X_o^2}$ . In a similar manner, given an initial condition of  $(X, Y, Z) = (-X_o, 0, 0)$ ,

X has a maximum of  $-X_o$  and a minimum of  $-\sqrt{4\sigma r + X_o^2}$ . Note that the average of  $X_{min}^2$  and  $X_{max}^2$  is equal to  $2\sigma r + X_o^2$ , which is equal to  $X_t^2$ .

Applying the initial conditions  $(X, Y, Z) = (\sqrt{2\sigma r}, 0, 0)$  that yield  $X_c = X_t = 2\sqrt{\sigma r} \ (\approx 31.62)$ , Figure 4 shows the closed form solutions obtained using Eq. (32), as well as numerical solutions of the 3D-NLM obtained using Eqs. (3-5). The results are in good agreement. For any positive  $X_o$ , a trajectory beginning with  $(X, Y, Z) = (X_o, 0, 0)$  is a closed curve. An oscillatory (closed) trajectory

associated with a center is orbitally stable (Jordan and Smith, 2007). In Sect. 3.4, the time varying energy cycle in Eq. (32) will be analyzed. Next, we discuss the solutions in (32) for the special case of  $X_o = 0$ .

- When considering a solution that begins with and ends at the saddle point, initial conditions of 335 (X,Y,Z) = (0,0,0) are required. At the saddle point, the time derivatives of the 3D-NLM in Eqs. (3-5) are zero. Therefore, numerically, only zero solutions for X, Y, and Z are possible. Obtaining a non-trivial solution can be attempted by initially adding a small perturbation (i.e., (X,Y,Z) = $(\epsilon,0,0)$ ) and then performing numerical integrations. A similar approach can be applied in Eq. (32) to obtain an approximate solution. Here, we provide an initial guess for  $\phi$ ,  $\phi(\tau) = \tau$ , over a target
- period in Eq. (32d). We then calculate X using Eq. (32c) and  $\phi$  using Eq. (32d) through iterations. Figure 5 displays the solutions determined using Eq. (32). The time evolution of Y is provided in Fig. 5a and suggests that Y increases slowly with time, reaches a maximum at  $\tau \approx 2.3$ , and then decreases with time to its minimum. The X-Y plot in Fig. 5b appears reasonable and is later compared to the analytical solution. In Fig. 5a, the evolution seems "periodic" but displays some irregularities that
- are different from those obtained using  $X_o \neq 0$ . This case is further analyzed using the following analytical solution, as well as the numerical solutions in Sect. 3.5.

From Eq. (C3) of Appendix C, the "second" half of the homoclinic orbital solution can be obtained as follows:

$$X(\tau) = \frac{4\sqrt{\sigma r}}{e^{\sqrt{\sigma r}\tau} + e^{-\sqrt{\sigma r}\tau}},$$
(34a)

350

$$Y(\tau) = -4r \frac{e^{\sqrt{\sigma r}\tau} - e^{-\sqrt{\sigma r}\tau}}{\left(e^{\sqrt{\sigma r}\tau} + e^{-\sqrt{\sigma r}\tau}\right)^2},\tag{34b}$$

$$Z(\tau) = \frac{X^2(\tau)}{2\sigma}.$$
(34c)

When t→∞, the above solution approaches the origin. Note that a special point on the solution
trajectory is (X,Y,Z) = (2√σr,0,2r). Therefore, the "second" half of the solution in Eq. (34) describes the trajectory that begins at (X,Y,Z) = (2√σr,0,2r) and that moves forward in time toward the origin, as shown in lighblue in Fig. 5b. The special point provides an alternative IC for the numerical integration of the 3D-NLM required to simulate (qualitatively) the homoclinic orbit, discussed with numerical results later. When t→∞, X(τ) ≈ 4√σre<sup>-√σrτ</sup> and Y(τ) ≈
-4re<sup>-√σrτ</sup>, yielding X/Y = -√σ/r. These features are the same as those for the linear (stable) solution with a heating term, as outlined in Sect. 3.1. As the system is invariant under t→ -t and y → -y (Strogatz, 2015), the solution (X(τ), -Y(τ), Z(τ)) in backward time, which represents the "firs