# Peer review of "On periodic solutions associated with the nonlinear feedback loop in the non-dissipative Lorenz model"

_Nonlinear Processes in Geophysics, 2016_

## Referee Comment (RC1) · Anonymous Referee #1 · 28 Sep 2016

The paper concerns itself with a set of equations that have its origins on the standard quasigeostrophic equations for atmospheric flows. The analysis is competent but not novel, the outcomes are not remarkable. Unless the Author could suggest in very specific and complete terms what relevance these equations have to a physical problem of interest it would be difficult for this paper to have a readership.

---

## Author Comment (AC1) · 29 Sep 2016

Thanks for your comments very much.

To facilitate discussions, the following quick responses are provided. More detailed discussions will be given in the final responses after the discussion period ends.

The Lorenz model (1963) has been studied extensively and been used to illustrate the sensitive dependence of solutions on initial conditions (i.e., the butterfly effect of the first kind.). Three types of physical processes in the Lorenz model are: heating, dissipation and nonlinear interactions.

Note that the nonlinearity is from the horizontal advection of temperature term, which appears in all of climate and weather models (e.g., Shen et al., 2006, 2012, 2013). Therefore, improving the understanding of the nonlinear term and the associated (thermodynamic) feedback may help improve the representation of the thermodynamic feedback in numerical models, which remains big uncertainties in climate model simulations.

In this study, we focus on the role of nonlinear processes and heating term (i.e., without the inclusion of dissipation). We present several closed-form solutions to the simplified Lorenz model. In addition to the closed-form solution using trigonometric functions, we also present a closed-form solution using elliptic functions in Appendix B. To our best knowledge, the solutions (to $X''+X^{**}3/2=0$) have not been documented before. The solutions can improve our understanding of the nonlinear processes and thus help examine the competing impact between the heating and nonlinearity.

Physically, we are able to relate the nonlinear term ($X^{**}3$) to the nonlinear feedback loop and show that the nonlinear feedback loop acts as a restoring force when a heating term (as well as dissipative term) is not included.

We then discuss how the collective impact of the nonlinear feedback loop and the heating may produce three types of solutions, including nonlinear periodic solutions with a small or big cycle and the homoclinic orbit solution, which are discussed in the manuscript. Note that further extensions of the nonlinear feedback loop and their collective impact with dissipations have been discussed in recent studies (Shen, 2014, 2015, 2016).

**References:**

Shen, B.-W., 2016: Hierarchical scale dependence associated with the extension of the nonlinear feedback loop in a seven-dimensional Lorenz model. Nonlin. Processes Geophys., 23, 189-203, doi:10.5194/npg-23-189-2016, 2016.
Shen, B.-W., 2015: Nonlinear Feedback in a Six-dimensional Lorenz Model. Impact of an additional heating term. Nonlin. Processes Geophys., 22, 749-764, doi:10.5194/npg-22-749-2015, 2015.

Shen, B.-W., 2014: Nonlinear Feedback in a Five-dimensional Lorenz Model. J. of Atmos. Sci., 71, 1701–1723. doi:http://dx.doi.org/10.1175/JAS-D-13-0223.1

Shen, B.-W., R. Atlas, O. Oreale, S.-J Lin, J.-D. Chern, J. Chang, C. Henze,and J.-L. Li, 2006: Hurricane Forecasts with a Global Mesoscale-Resolving Model: Preliminary Results with Hurricane Katrina(2005). Geophys. Res. Lett., L13813, doi:10.1029/2006GL026143.

Shen, B.-W. W.-K. Tao, Y.-L. Lin, A. Laing, 2012: Genesis of Twin Tropical Cyclones as Revealed by a Global Mesoscale Model: The Role of Mixed Rossby Gravity Waves. J. Geophys. Res., 117, D13114, doi:10.1029/2012JD017450.

Shen, B.-W., M. DeMaria, J.-L. F. Li, and S. Cheung, 2013: Genesis of Hurricane Sandy (2012) Simulated with a Global Mesoscale Model. Geophys." Res. Lett. 40. 2013, DOI: 10.1002/grl.50934.

---

## Referee Comment (RC2) · Anonymous Referee #2 · 5 Dec 2016

This manuscript discusses a model called by the author the non-dissipative Lorenz model. The model is related to the Lorenz 1963 model, but with several terms missing. It is apparently derived from the Boussinesq equations by a method similar to that of Lorenz. The model has a conserved quantity, and for a particular value of the conserved quantity, the model is equivalent to the Duffing equation with no damping and no forcing, i.e., frictionless motion in a double-well potential given by a symmetric 4th-degree polynomial. The analysis of periodic and homoclinic solutions of this model is not novel, and it is not clear to me that it offers significant insight into the physics of the Boussinesq equations.

To be more specific, equation (15) in the manuscript is the Duffing equation discussed above; I find it peculiar that the manuscript does not mention Duffing until Appendix B. The Duffing equation has been studied extensively, and in the particular case of no

damping and no forcing, its solutions are particularly well understood. The closed-form solution in the special case presented in Appendix B is also derived, for example, in the discussion below equation (23) of the article at:

http://www.sciencedirect.com/science/article/pii/S0094114X14002079

Closed-form solutions for more general equations are derived in the articles at:

http://www.sciencedirect.com/science/article/pii/S0307904X12002302         http://isi-dl.com/downloadfile/106810

The present manuscript focuses primarily on the analysis of the Duffing model, which I already well understand, and I don't feel that I learned anything from it about the Lorenz or Boussinesq equations.

---

## Author Comment (AC2) · 6 Dec 2016

Thanks for your comments very much.

To facilitate discussions, the following quick responses are provided. More detailed discussions will be given in the final responses after the discussion period ends.

We are aware that a further simplified 3D-NLM (e.g., with a particular set of initial conditions) and the Duffing equation (as discussed in the Appendix) (or double-well potential system) may be dynamically equivalent. However, while the former (the 3D-NLM) has three ordinary differential equations (ODEs) and the latter (with or without an external forcing) is a second-order ODE. Therefore, without providing a proof regarding a homeomorphism, we avoided a detailed comparison between the two systems. While our focus of this study is on the role of the nonlinear feedback loop in producing oscillatory solutions (and the homoclinic orbit), our new paper is to examine the impact of the extended nonlinear feedback loop on the periodicity (or quasi-periodicity) of solutions in a five-dimensional non-dissipative Lorenz model, which is simplified from the five-dimensional Lorenz model (Shen, 2014). Here, we would like to present how the nonlinear feedback loop (i.e., -XZ and XY) may lead to the nonlinear restoring forcing term (i.e., $X^{**}3$), which plays a role in producing oscillatory solutions.

The Lorenz model (1963) has been studied extensively and been used to illustrate the sensitive dependence of solutions on initial conditions (i.e., the butterfly effect of the first kind.). Three types of physical processes in the Lorenz model are: heating, dissipation and nonlinear interactions. The nonlinearity is from the horizontal advection of temperature term, which appears in all of climate and weather models (e.g., Shen et al., 2006, 2012, 2013). Therefore, improving the understanding of the nonlinear term and the associated (thermodynamic) feedback may help improve the representation of the thermodynamic feedback in numerical models, which remains big uncertainties in climate model simulations.

In this study, we focus on the role of nonlinear processes and heating term (i.e., without the inclusion of dissipation). We present several closed-form solutions to the simplified Lorenz model. In addition to the closed-form solution using trigonometric functions, we also present a closed-form solution using elliptic functions in Appendix B as verifications. It is our believe the simple form of the solutions (to $X''+X^{**}3/2=0$) using trigonometric functions can effectively help illustrate the role of the nonlinear term ($X^{**}3$) in producing oscillatory solutions. The solutions can improve our understanding of the nonlinear processes and thus help examine the competing impact between the heating and nonlinearity. The relationship between the nonlinear feedback loop (-XZ and XY) and the nonlinear term $X^{**}3$ is given on page 3 of the response file. The physical processes (i.e., downscaling and upscaling processes) associated with -XZ and XY were first discussed in Shen (2014).

Additionally, we discuss how the collective impact of the nonlinear feedback loop and the heating may produce three types of solutions, including nonlinear periodic solutions with a small or large cycle and the homoclinic orbit solution, and discuss the energy cycle. Note that $X^{**}2$, $Y^{**}2+Z^{**}2$ and $Z$ are associated with kinetic, available potential

and potential energy, respectively. (This should be different from the energy cycle in the Duffing equation). As mentioned, we currently examine the impact of the extended nonlinear feedback loop on the periodicity (or quasi-periodicity) of solutions using a five-dimensional non-dissipative Lorenz model. More detailed responses will be given soon.

**References:**

Shen, B.-W., 2016: Hierarchical scale dependence associated with the extension of the  nonlinear feedback loop in a seven-dimensional Lorenz model. Nonlin. Processes Geophys., 23, 189-203, doi:10.5194/npg-23-189-2016, 2016.

Shen, B.-W., 2015: Nonlinear Feedback in a Six-dimensional Lorenz Model. Impact of an additional heating term. Nonlin. Processes Geophys., 22, 749-764, doi:10.5194/npg-22-749-2015, 2015.

Shen, B.-W., 2014: Nonlinear Feedback in a Five-dimensional Lorenz Model. J. of Atmos. Sci., 71, 1701–1723. doi:http://dx.doi.org/10.1175/JAS-D-13-0223.1

Shen, B.-W., R. Atlas, O. Oreale, S.-J Lin, J.-D. Chern, J. Chang, C. Henze,and J.-L. Li, 2006: Hurricane Forecasts with a Global Mesoscale-Resolving Model: Preliminary Results with Hurricane Katrina(2005). Geophys. Res. Lett., L13813, doi:10.1029/2006GL026143.

Shen, B.-W. W.-K. Tao, Y.-L. Lin, A. Laing, 2012: Genesis of Twin Tropical Cyclones as Revealed by a Global Mesoscale Model: The Role of Mixed Rossby Gravity Waves. J. Geophys. Res., 117, D13114, doi:10.1029/2012JD017450.

Shen, B.-W., M. DeMaria, J.-L. F. Li, and S. Cheung, 2013: Genesis of Hurricane Sandy (2012) Simulated with a Global Mesoscale Model. Geophys." Res. Lett. 40. 2013, DOI: 10.1002/grl.50934.

The non-dissipative Lorenz model (3D-NLM) with no heating term is written as:

$$\frac{\mathrm{d}X}{\mathrm{d}\tau} = \sigma Y, \tag{1}$$

$$\frac{\mathrm{d}Y}{\mathrm{d}\tau} = -XZ, \tag{2}$$

$$\frac{\mathrm{d}Z}{\mathrm{d}\tau} = XY. \tag{3}$$

Equations (1-2) lead to

$$\frac{d^2X}{d\tau^2} = \sigma\frac{dY}{d\tau} = -\sigma XZ. \tag{4}$$

From Eqs. (1) and (3) (i.e., $X * Eq.(1) - \sigma * Eq.(3)$), we have

$$\frac{d}{d\tau}\left(\frac{X^2}{2} - \sigma Z\right) = 0, \tag{5}$$

and

$$\frac{X^2}{2} - \sigma Z = C, \tag{6}$$

where $C$ is a constant. The initial condition of $(X, Y, Z) = (0, 1, 0)$ leads to $C = 0$. Therefore, Eqs. (4) and (5) with the above initial condition gives us

$$\frac{d^2X}{d\tau^2} = -\frac{X^3}{2}.$$

The above derivations illustrate the relationship between the $X^3$ and the nonlinear feedback loop (i.e., $-XZ$ and $XY$), which are associated a pair of upscaling and downscaling processes as discussed in Shen (2014).

---

## Author Comment (AC3) · 14 Dec 2016

**B.-W. Shen**

bshen@mail.sdsu.edu

In our recent responses to reviewer II's comments, we mentioned that the current study provides a baseline for further examining the role of the extended nonlinear feedback loop in producing quasi-periodic solutions in a 5D non-dissipative Lorenz model. Here, we would like to share some of preliminary results which were presented by Ms. Faghih-Naini on December 8. The presentation slides are uploaded as supplemental materials. Based on the results, a manuscript is in preparation and will be submitted for publication soon.

Citation: Faghih-Naini, S. and B.-W. Shen, 2016: Quasi-periodic solutions associated with the extended nonlinear feedback loop of the five-dimensional non-dissipative Lorenz model. Project presentation by Ms. Faghih-Naini. SDSU GMCS 422. December 08, 2016. (A manuscript is in preparation).

Please also note the supplement to this comment:
http://www.nonlin-processes-geophys-discuss.net/npg-2016-40/npg-2016-40-AC3-supplement.pdf

[Figure]

**Supplement:**

**Quasi-periodic solutions associated with the extended nonlinear feedback loop of the five-dimensional nondissipative Lorenz model**

Sara Faghih-Naini

Math596 - High Performance Computing For Applied Mathematics
Mathematics and Statistics
San Diego State University

12/08/2016

together with Dr. Bo-Wen Shen

**Outline**

**On quasi-periodic solutions associated with the extended nonlinear feedback loop of the five-dimensional nondissipative Lorenz model**

Sara Faghih-Naini, San Diego State University

**Objective**

- **Analysis and comparison of 5D NLM**
- **Fundamental role of nonlinear terms in producing quasi-periodic solutions**
- **Collective role of the nonlinear feedback loop and heating term in producing quasi-periodic solutions**

[Figure]

2D view          3D view

Y-Y1-plots of periodic/quasiperiodic solutions for tend=0.512 (top) and tend=5.12 (bottom) of coupled (left) and uncoupled (right) LL 5D NLM

**Approach**

- **Performing analytical, symbolic (python scipy) and numerical (Python ODE solver (odeint)) analysis of 5D NLM**
- **Analyzing quasiperiodic solutions of 5D NLM**
- **Comparing to solutions of 5D NLM, locally linearized 5D NLM, 3D NLM, …**
- **Performing frequency analysis using Python fft package**
- **Visualizing quasiperiodic solutions using Python visualization packages**

**Key Milestones**

- **09/30/2016: solve for eigenvalues of 5D NLM (analytical and symbolic computation)**
- **11/01/2016: verify results, plots for comparison of 3D NLM, LL 5D NLM FN=0 and FN=1, 3D-eigenvaluesolution (analytical and symbolic); plots for comparison of 5D NLM, LL 5D NLM FN=0 and FN=1, 5D-eigenvaluesolution (analytical and symbolic)**
- **11/20/2016: finish frequency analysis**
- **11/22/2016: create slides for presentation**
- **11/30/2016: finish oral presentation**
- **12/06/2016: draft of paper**

Math596_HPC@SDSU

Figure: QuadChart

**Introduction: Quasiperiodic Solutions**

- ▶ Frequency ratio $\frac{\omega_2}{\omega_1}$ of two modes determines behavior of solution:
- ▶ If $\frac{\omega_2}{\omega_1}$ is rational, the motion is periodic and has a closed orbit.
- ▶ If $\frac{\omega_2}{\omega_1}$ is irrational, these two frequencies are called incommensurate;
- ▶ The composite motion is quasiperiodic and its period is infinite;
- ▶ The trajectory is dense, that means, it comes arbitrarily close to each point of torus.

**Introduction: Quasiperiodic Solutions**

[Figure]

Figure: Quasiperiodic solution of coupled LL 5D NLM with a frequency ratio of $\frac{1}{2}\left(3-\sqrt{5}\right)$ (top) and periodic solution of uncoupled LL 5D NLM with a frequency ratio of $2$ (bottom) up to time 0.512 (left) and 5.12 (middle)

**5D NLM V1**

Removing the dissipative terms from the five dimensional Lorenz model (5D LM), the the nondisipative five dimensional Lorenz model (5D NLM) results:

$$\frac{dX}{d\tau} = -\sigma X + \sigma Y, \tag{1}$$

$$\frac{dY}{d\tau} = -XZ + rX - Y, \tag{2}$$

$$\frac{dZ}{d\tau} = XY \boxed{-XY_1} - bZ, \tag{3}$$

$$\frac{dY_1}{d\tau} = \boxed{XZ} - 2XZ_1 - d_0 Y_1, \tag{4}$$

$$\frac{dZ_1}{d\tau} = 2XY_1 - 4bZ_1. \tag{5}$$

**5D NLM V2**

Perturbation method leads to the locally linearized 5D NLM (LL 5D NLM):

$$\frac{dX'}{d\tau} = \sigma Y', \tag{6}$$

$$\frac{dY'}{d\tau} = (r - Z_c)X' - X_cZ' - FN(X'Z'), \tag{7}$$

$$\frac{dZ'}{d\tau} = (Y_c - Y_{1c})X' + X_cY' - \boxed{X_cY_1'} + FN(X'Y' - X'Y_1'), \tag{8}$$

$$\frac{dY_1'}{d\tau} = (Z_c - 2Z_{1c})X' + \boxed{X_cZ'} - 2X_cZ_1' + FN(X'Z' - 2X'Z_1'), \tag{9}$$

$$\frac{dZ_1'}{d\tau} = 2Y_{1c}X' + 2X_cY_1' + 2FN(X'Y_1'). \tag{10}$$

Choosing $FN = 0$ makes the system linear with respect to the critical point and for $FN = 1$ the system is fully nonlinear.

**Model Assumptions and List of Simulations**

Parameters: $\sigma = 10, r = 25, \Delta\tau = 0.001$

| Method | Model | Equations | IC | Python Packages |
|--------|-------|-----------|-----|-----------------|
| analytical | V2 FN=0 | 6-10 | $X_0 = \sqrt{\frac{5}{2}\sigma r}$, $X_c = \sqrt{\frac{5}{2}\sigma r + X_0^2}$ | - |
| symbolic | V2 FN=0 | 6-10 | $X_0 = \sqrt{\frac{5}{2}\sigma r}$, $X_c = \sqrt{\frac{5}{2}\sigma r + X_0^2}$ | linalg |
| numerical | V2 FN=0 V2 FN=1 V1 | 6-10 6-10 1-5 | $X_c = \sqrt{\frac{5}{2}\sigma r + X_0^2}$ for $X_0 = 0.25\sqrt{\frac{5}{2}\sigma r}$, $X_0 = 0.5\sqrt{\frac{5}{2}\sigma r}$, $X_0 = \sqrt{\frac{5}{2}\sigma r}$, $X_0 = 4\sqrt{\frac{5}{2}\sigma r}$, | odeint |

Impact of ICs

Table: Methods, corresponding model assumptions and computing packages used for verifying results and analyzing 5D NLM

**Analytical Solution for Eigenvalues**

Setting $FN = 0$, plugging in the basic state values, the Jacobian matrix becomes:

$$A^{5DNLM} = \begin{pmatrix} 0 & \sigma & 0 & 0 & 0 \\ 0 & 0 & -X_c & 0 & 0 \\ 0 & X_c & 0 & \boxed{-X_c} & 0 \\ 0 & 0 & \boxed{X_c} & 0 & -2X_c \\ 0 & 0 & 0 & 2X_c & 0 \end{pmatrix}$$

To obtain the eigenvalues $\lambda_1, ... \lambda_5$, with $\lambda = i\beta$, the equation $det(A^{5DNLM} - i\beta\mathbb{I}) = 0$ was solved applying Laplace's formula:

$$\beta_1 = \sqrt{\left(3 + \sqrt{5}\right)}X_c$$

$$\beta_2 = -\sqrt{\left(3 + \sqrt{5}\right)}X_c$$

$$\beta_3 = \sqrt{\left(3 - \sqrt{5}\right)}X_c$$

$$\beta_4 = -\sqrt{\left(3 - \sqrt{5}\right)}X_c$$

$$\beta_5 = 0$$

**Analytical Solution for Eigenvectors**

For $k = 1, 2, 3, 4$ the form of the corresponding eigenvectors is

$$v_k = \begin{pmatrix} \sigma \left( \frac{-\beta_k^2}{2X_c^3} + \frac{5}{2X_c} \right) \\ i\beta_k \left( \frac{-\beta_k^2}{2X_c^3} + \frac{5}{2X_c} \right) \\ \frac{-\beta_k^2}{2X_c^2} + 2 \\ \frac{i\beta_k}{2X_c} \\ 1 \end{pmatrix} \text{ and } v_5 = \begin{pmatrix} \sigma \\ 0 \\ 0 \\ 0 \\ 0 \end{pmatrix}.$$

This leads to the general solution
$S(t) = C_1 e^{i\beta_1 t} v_1 + C_2 e^{i\beta_2 t} v_2 + C_3 e^{i\beta_3 t} v_3 + C_4 e^{i\beta_4 t} v_4 + C_5 e^{i\beta_5 t} v_5$, which can be expressed in the sine-cosine representation.

**Analysis of Analytical Solution**

- $C = C_1, ..., C_5$ can be determined by applying the initial condition $I = (X_0 - X_c, 0, -r, 0, -\frac{r}{2})$.

- Using the analytical solutions of the eigenvectors, which include oscillatory modes with two incommensurate frequencies, as a basis for a solution, the coefficients of the eigenvectors can be determined as $C_1 + C_2 = \frac{-\beta_3{}^2 r}{2(\beta_3{}^2 - \beta_1{}^2)}$,

  $C_3 + C_4 = \frac{\beta_1{}^2 r}{2(\beta_3{}^2 - \beta_1{}^2)}$ and $C_5 = \frac{X_0 - X_c}{\sigma} + \frac{5r}{4X_c}$ (constant mode) (for more detailed discussions see Appendix)

- The **frequency ratio** $\frac{\beta_3}{\beta_1} = \frac{\sqrt{(3-\sqrt{5})}X_c}{\sqrt{(3+\sqrt{5})}X_c} = \frac{1}{2}\left(3 - \sqrt{5}\right)$ is **irrational**, so solution is **quasiperiodic**.

[Figure]

**Analytical vs. Symbolic Solution-shorter time**

$X_0 = \sqrt{\frac{5}{2}\sigma r}, X_c = \sqrt{\frac{5}{2}\sigma r + X_0^2}$

analytical and symbolic solutions are identical

Figure: Analytical vs. Symbolic Solutions for $X_0 = \sqrt{\frac{5}{2}\sigma r}$, $X_c = \sqrt{\frac{5}{2}\sigma r + X_0^2}$ and $\tau \in [0, \mathbf{0.512}]$

[Figure]

**Analytical vs. Symbolic Solution-longer time**

$$X_0 = \sqrt{\tfrac{5}{2}\sigma r}, X_c = \sqrt{\tfrac{5}{2}\sigma r + X_0^2}$$

analytical and symbolic solutions are identical

Figure: Analytical vs. Symbolic Solutions for $X_0 = \sqrt{\frac{5}{2}\sigma r}$, $X_c = \sqrt{\frac{5}{2}\sigma r + X_0^2}$ and $\tau \in [0, \mathbf{1.024}]$

**Model Verification**

$X_0 = \sqrt{\frac{5}{2}\sigma r}, X_c = \sqrt{\frac{5}{2}\sigma r + X_0^2}$

analytical and numerical solutions are identical

Figure: Analytical vs. Numerical solution of LL 5D NLM (V2, FN=0) for $X_0 = \sqrt{\frac{5}{2}\sigma r}$, $X_c = \sqrt{\frac{5}{2}\sigma r + X_0^2}$ and $\tau \in [0, 0.512]$

**Impact of Nonlinearity**

[Figure]

Figure: Numerical solutions of LL 5D NLM FN=0 vs. FN=1 for
$X_0 = \sqrt{\frac{5}{2}\sigma r}$, $X_c = \sqrt{\frac{5}{2}\sigma r + X_0^2}$ and $\tau \in [0, 0.512]$

**Impact of Initial Conditions**

$X_c = \sqrt{\frac{1}{2}\sigma r + X_4^2}$ V2 with different ICs

[Figure]

Figure: Comparison of different initial conditions in LL 5D NLM FN=0 and FN=1 for $t \in [0, 0.512]$

**Examine coupling terms using 5D NLM V2**

$$\frac{dX'}{d\tau} = \sigma Y',$$

$$\frac{dY'}{d\tau} = (r - Z_c)X' - X_cZ' - FN(X'Z'),$$

$$\frac{dZ'}{d\tau} = (Y_c - Y_{1c})X' + X_cY' - \boxed{X_cY_1'}^{\,1} + FN(X'Y' - X'Y_1'),$$

$$\frac{dY_1'}{d\tau} = (Z_c - 2Z_{1c})X' + \boxed{X_cZ'}^{\,2} - 2X_cZ_1' + FN(X'Z' - 2X'Z_1'),$$

$$\frac{dZ_1'}{d\tau} = 2Y_{1c}X' + 2X_cY_1' + 2FN(X'Y_1').$$

[Figure]

**Impact of coupling terms**

[Figure]

Figure: Solution of LL 5D NLM FN=0 becomes peridoic, if coupling terms $X_c Y_1$ in $\frac{dZ'}{d\tau}$ and/or $X_c Z$ in $\frac{dY_1'}{d\tau}$ are ignored

**Summary**

▶ For the locally linear 5D NLM, its analytical solution with two
  incommensurate frequencies, whose ratio is irrational, was
  obtained.

▶ A comparison between the 3D NLM and 5D NLM suggests
  that the incommensurate frequencies in the 5D NLM are
  produced by the extension of the nonlinear feedback loop.

▶ While the 3D NLM includes periodic solutions, the 5D NLM
  produces a quasi-periodic solution. The coupling terms that
  are associated with the extension of the nonlinear feedback
  loop are crucial for the appearance of incommensurate
  frequencies in the 5D NLM.

▶ Linear and nonlinear solutions for the 5D NLM (i.e., version 2)
  are closer for greater values for $X_c$, i.e. values being further
  away from the origin.

[Figure]

**Appendix**
**Analytical Solution - General Form**

$C = C_1, ..., C_5$ can be determined by setting $t = 0$ and equating $S$ with the initial condition $I = (X_0 - X_c, 0, -r, 0, -\frac{r}{2})$. The the system has the following form:

$$S(t) = \frac{-\beta_3{}^2 r}{2(\beta_3{}^2 - \beta_1{}^2)} \begin{pmatrix} \sigma\left(\frac{-\beta_1{}^2}{2X_c^3} + \frac{5}{2X_c}\right)cos(\beta_1 t) \\ -\beta_1\left(\frac{-\beta_1{}^2}{2X_c^3} + \frac{5}{2X_c}\right)sin(\beta_1 t) \\ \left(\frac{-\beta_1{}^2}{2X_c^2} + 2\right)cos(\beta_1 t) \\ \frac{-\beta_1}{2X_c}sin(\beta_1 t) \\ cos(\beta_1 t) \end{pmatrix} +$$

$$\frac{\beta_1{}^2 r}{2(\beta_3{}^2 - \beta_1{}^2)} \begin{pmatrix} \sigma\left(\frac{-\beta_3{}^2}{2X_c^3} + \frac{5}{2X_c}\right)cos(\beta_3 t) \\ -\beta_3\left(\frac{-\beta_3{}^2}{2X_c^3} + \frac{5}{2X_c}\right)sin(\beta_3 t) \\ \left(\frac{-\beta_3{}^2}{2X_c^2} + 2\right)cos(\beta_3 t) \\ \frac{-\beta_3}{2X_c}sin(\beta_3 t) \\ cos(\beta_3 t) \end{pmatrix} + \left(\frac{X_0 - X_c}{\sigma} + \frac{5r}{4X_c}\right)\begin{pmatrix} \sigma \\ 0 \\ 0 \\ 0 \\ 0 \end{pmatrix}$$

[Figure]

**Symbolic Solution**

```python
from sympy import Symbol, Matrix

X_c = Symbol('X_c')
sig = Symbol('sig')
la= Symbol('la')
A5D=Matrix([[0,sig,0,0,0],
            [0,0,-X_c,0,0],
            [0,X_c,0,-X_c,0],
            [0,0,X_c,0,-2*X_c],
            [0,0,0,2*X_c,0]])
eigenvalues=A5D.eigenvals()
eigenvectors=A5D.eigenvects()
```

Figure: Python code for symbolic calculation of eigenvalues and eigenvectors

**Numerical Solution - 5D NLM V1**

```python
from scipy.integrate import odeint
import numpy as np

def Lorenz5D_nondissipative(state,t, sigma, r, beta, d0):

    x,y,z,y1,z1 = state

    dx = sigma * (y)
    dy = -x*z   + r*x
    dz =  x*y   - x*y1
    dy1 = x*z   - 2*x*z1
    dz1 = 2*x*y1

    return [dx, dy, dz, dy1, dz1]

sigma = 10.0
r = 25.0
beta = 8.0/3.0
d0 = 19.0/3.0

X0=np.sqrt(5./2*sigma*r)
xc=np.sqrt(5./2*sigma*r+X0**2)
zc=r
z1c = r/2
yc= 0.
y1c= 0.

dt=0.001
L=512*dt
t = np.arange(0.0,L, dt)
state = [X0_1, 0, 0, 0, 0]

out = odeint(Lorenz5D_nondissipative, state, t,  args=(sigma, r, beta, d0))
```

Figure: Python code for numerical solution of the 5D NLM using odeint

**Numerical Solution - 5D NLM V2**

```python
from scipy.integrate import odeint
import numpy as np

def LL_Lorenz5D(state, t, parameters):

    x, y, z, y1, z1 = state
    sigma, r, beta, d0, FN, X0, xc, zc, yc, z1c, y1c = parameters

    dx = sigma * (y)
    dy = (r-zc)*x  - xc*z - FN*(x*z)
    dz = (yc-y1c)* x + xc*y - xc*y1 + FN*(x*y-x*y1)
    dy1 = (zc-2*z1c)*x + xc * z  - 2*xc*z1 + FN*(x*z-2*x*z1)
    dz1 = 2*y1c*x + 2*xc*y1 + 2*FN*(x*y1)

    return [dx, dy, dz, dy1, dz1]

sigma = 10.0
r = 25.0
beta = 8.0/3.0
d0 = 19.0/3.0

X0=np.sqrt(5./2*sigma*r)
xc=np.sqrt(5./2*sigma*r+X0**2)
zc=r
z1c = r/2
yc= 0.
y1c= 0.

dt=0.001
L=512*dt
t = np.arange(0.0, L, dt)
state = [X0_1-xc_1, 0, -r, 0, -r/2] #=[X', Y', Z', Y1', Y2']

FN=0
parameters = (sigma, r, beta, d0, FN, X0, xc, zc, yc, z1c, y1c)
out0 = odeint(LL_Lorenz5D, state, t, args=(parameters,))
FN=1
parameters = (sigma, r, beta, d0, FN, X0, xc, zc, yc, z1c, y1c)
out1 = odeint(LL_Lorenz5D, state, t, args=(parameters,))
```

Figure: Python code for numerical solution of the LL 5D NLM using odeint.

**Frequency analysis for LL 5D NLM**

V2 frequency analysis $X_0 = \sqrt{\frac{1}{2}\sigma r}$, $X_c = \sqrt{\frac{1}{2}\sigma r + X_0^2}$ magnitudes

Figure: Magnitudes of fourier modes of LL 5D NLM FN=0 and FN=1 and zoomed-in presentation for comparison with eigenvalue solution for $\tau \in [0, 0.512]$

---

## Author Comment (AC4) · 23 Feb 2017

Dear Editor:

Please consider for publication on *nonlinear processes in geophysics*

the following re-revised and resubmitted manuscript

``

On the nonlinear feedback loop and energy cycle of the non-dissipative Lorenz model

''

by Bo-Wen Shen

In this study using the 3D-NLM ([npg-16-40](npg-16-40)) and a new manuscript using the 5D-NLM (Faghih-Naini and Shen, 2017, submitted, [npg-17-2](npg-17-2)), we would like to illustrate the fundamental role of the nonlinearity in producing "recurrent'' solutions, including periodic orbits within the 3D-NLM and quasi-periodic orbits within the 5D-NLM. Here, the nonlinearity can be identified as the nonlinear feedback loop and its extension. Since the original manuscript was published as a discussion article in April 2014, our goal has been to illustrate the role of the nonlinear feedback loop (-XZ and XY) in producing a periodic solution using a very simple nonlinear ordinary differential equation (ODE), $X''+X^3/2=0$, where the nonlinear term $X^3$ comes from the nonlinear feedback loop. The abstract in npg-16-40 does indicate this. After the 2014 discussion article being reviewed, we incrementally added other parts, including a comparison of the model with the Duffing equation (e.g., Appendix B) into the npg-16-40. On the other hand, compared to the Duffing equation, the 3D-NLM was used to "indicate" the importance of applying adaptive methods for the numerical integration of the homoclinic orbit in the 3D phase space (e.g., Figs. 7c-d). We also discussed the partitions of the averaged available potential energy from Y and Z modes. Note that the Diffing equation does not include a component that is similar to the Z mode. We believe that our main points in npg-16-40 are valuable and have not been discussed in the literatures. Additionally, results using the 3D-NLM laid the foundation for the recent work using the 5D-NLM in npg-17-2. While detailed responses are given in the sections of "general responses'' and "specific responses'', I would like to provide the following responses below.

We did state that the 3D-NLM with certain types of ICs can be reduced to become the Duffing equation (e.g., Appendix B in npg-16-40 or Appendix C in the revised

manuscript), which can help verify the periodicity of solutions and thus build our confidence in the role of nonlinear feedback loop. Additionally, in the newly added Appendix B, we discussed the relationship between the 3D-NLM with r=0 and the 3DLM with a very large r (e.g., Sparrow, 1982), and the real-valued Maxwell-Bloth equations (David and Holm, 1992). All of the three systems can be reduced to become $X''+X^3/2$. Therefore, based on our analysis in npg-16-40 and the 2014 discussion article, two nonlinear terms, -XZ and XY, also acts as a nonlinear restoring forcing in the $2^{nd}$ system (i.e., the 3DLM with a large r); and two nonlinear terms, XZ and –XY, in the $3^{rd}$ system (i.e., i.e., the real-valued Maxwell-Bloth equations) play a similar role. Under a certain range of initial conditions, these system and their solutions may be comparable to a simplified Duffing equation (e.g., Appendix C and Roupas, 2012). However, we want to emphasize that our goal is to discuss the role of nonlinear feedback loop in producing periodic (or recurrent) solutions using the 3D-NLM (as well as in producing quasi-periodic solutions using the 5D-NLM).

By comparison, Appendix C of Faghih-Naini and Shen (2017, npg-17-2) has provided an analogy between the locally linear 5D-NLM and a coupled system with two springs that both systems have the same mathematical equations when specific spring constants are selected. This kind of comparison is consistent with our approach of comparing the 3D-NLM, the Duffing equation, the 3DLM with a large r and the real-valued Maxwell-Bloth equations. Through these comparisons, we can build our confidence in the accuracy of mathematical solutions. Then, we could use the results (both solutions and the corresponding terms, including –XZ and XY) to improve our understanding of the underlying physical processes in the specific system. In our papers with the 3D-NLM (npg-16-40), 5D-NLM (npg-17-2) and 7D-NLM (Shen and Faghih-Naini, 2017), we illustrated the role of the nonlinear feedback loop and its extension in producing recurrent (i.e., periodic or quasi-periodic) solutions. As the nonlinear feedback loop appears throughout the spatial mode-mode interactions rooted in the nonlinear temperature advection, we believe that the nonlinear feedback loop can also produce quasi-periodic solutions in nonlinear numerical models based on Navier-Stokes equations.

In a brief summary, in the revised manuscript, our main focus is to illustrate that the nonlinear feedback loop serves as the nonlinear restoring force using closed solutions of a

very simple nonlinear ODE, $X''+X^3/2=0$, and numerical solutions of the 3D-NLM (e.g., Eqs. 3-5 with r=0 and r≠0). The study in npg-16-40 has been extended to show the role of an extended nonlinear feedback loop in producing recurrent (quasi-periodic) solutions using 5D-NLM (npg-16-40 by Faghih-Naini and Shen, 2017) as well as 7D-NLM (Shen and Faghih-Naini, 2017). These non-dissipative LMs will be compared with the corresponding dissipative LMs to understand the impact of dissipations and their interaction with the non-linear feedback loop on chaotic solutions (e.g., topological transitivity).

Detailed responses are provided below. A pdf file that includes responses to the comments on the 2014 discussion article is also attached. A revised manuscript with changes highlighted in red is uploaded. We really appreciate reviewers' and Editor's comments that have greatly improved the quality of the manuscript. We thank the reviewers and Editor for providing us the opportunity for explaining further the progress We have made and hope that my responses are acceptable. Thank you for your consideration!

-Bowen

Associate Professor
Department of Mathematics and Statistics
San Diego State University
5500 Campanile Drive
San Diego, CA 92182-7720
Tel: 619-594-5962
Email: bshen@mail.sdsu.edu
URL: http://bwshen.sdsu.edu

*
* General Responses:
*

The Lorenz model (1963) has been studied extensively and been used to illustrate the sensitive dependence of solutions on initial conditions (i.e., the butterfly effect of the first kind.). *Three types of physical processes in the Lorenz model are: heating, dissipation and nonlinear interactions.* Our studies in npg-16-40 and other papers (e.g., Shen, 2014, 2015, 2016) have been performed to understand their individual and/or collective impact on the following characteristics of a chaotic system defined by Devaney (1989): (1) sensitivity to initial conditions; (2) topological transitivity; and (3) dense periodic points. As the $3^{rd}$ feature suggests "recurrence", and our analysis (Shen 2014) indicated a nonlinear feedback loop in the 3D Lorenz model (3DLM) as well as 5DLM, we have analyzed the 3D non-dissipative LM (3D-NLM) and high-dimensional non-dissipative LMs to understand the relationship between recurrent solutions (i.e., periodic of quasi-periodic solutions) and nonlinear feedback loop. [Here, a recurrence is defined when the distance between two states at different times within the phase space is smaller than a threshold ε. Mathematically, the recurrence may be associated with non-zero imaginary parts of the eigenvalues in the locally linear system near a non-trivial critical point.]

The nonlinearity (i.e., the nonlinear feedback loop) in the 3DLM is from the horizontal advection of temperature term (e.g., Shen, 2014), which appears in all of climate and weather models (e.g., Shen et al., 2006, 2012, 2013). Therefore, improving the understanding of the nonlinear terms and the associated (thermodynamic) feedback may help improve the representation of the thermodynamic feedback in numerical models, which remains big uncertainties in climate model simulations. While the role of the nonlinear feedback loop has been discussed using the original and high-dimensional Lorenz models (Shen, 2014, 2015, 2016), it is our belief that the role of nonlinearity can be better understood using non-dissipative versions, as discussed in this study (npg-16-40) and recent studies by Faghih-Naini and Shen (2017, npg-17-2) and Shen and Faghih-Naini (2017).

Using the 3D-NLM, we have two discussion articles posted by NPGD, which are referred to as the npg-14-21 and npg-16-40, respectively. The npg-16-40 discussion article was expanded based on the comments of the reviewers on the npg-14-21

discussion article. Recently, the study with the 3D-NLM has been extended using the 5D-NLM (Faghih-Naini and Shen, 2017, npg-17-2). In the studies with the 3D-NLM and 5D-NLM, we discussed the fundamental role of the nonlinearity (represented by the nonlinear feedback loop in our papers) in producing ``recurrent'' solutions, including periodic orbits within the 3D-NLM and quasi-periodic orbits within the 5D-NLM. The "simplified'' non-dissipative systems are used to reveal the relationship between the nonlinear feedback loop and the "recurrence" (i.e., periodicity or quasi-periodicity, to be specific). [Note that a future study is to compare the results with non-dissipative and dissipative models to analyze topological transitivity.]

The main focus in both the npg-14-21 and npg-16-40 discussion articles has been on the closed-form solution of the simple nonlinear ODE, $X''+X^3/2 = 0$, and the role of the nonlinear term $X^3$. It is our believe the simplicity of the solutions (to $X''+X^3/2=0$) using trigonometric functions can effectively help illustrate the role of the nonlinear term ($X^3$) in producing oscillatory solutions. The cubic nonlinear term comes from the nonlinear feedback loop ($-XZ$ and $XY$), which is briefly discussed in Appendix B in the revised manuscript, and the nonlinear feedback loop is from the nonlinear advection temperature as first discussed in Shen (2014). To our best knowledge, our closed-form solution to the above equation has never been documented in the literature. Additionally, we discussed the energy cycle associated with the periodic solutions using the 3D-NLM where evolution of kinetic, potential energy and available potential energy can be discussed. [Note that if we simply consider $X''+X^3/2=0$, only kinetic and potential energy can be defined.] The solution (to $X''+X^3/2=0$) can improve our understanding of the nonlinear processes and thus help examine the competing impact between the heating and nonlinearity. In the npg-16-40 discussion article, the other solutions (e.g., solutions using elliptic functions) were added partially in response to reviewers comments and suggestions. The solutions were used to support the view that the nonlinear feedback loop (with or without heating) can produce oscillatory solutions in the 3D-NLM.

When the relationship between the 3D-NLM, $X''+X^3/2=0$, and the Duffing equation was discussed in Appendix B of npg-16-40 article or Appendix C of the revised manuscript. Per reviewer's comments, we added related discussion in the main text of the revised manuscript as well. Additionally, we added a new Appendix B to discuss the

relationship between the 3D-MLM with r=0 and the other two systems, including the original 3DLM with a large r (Sparrow, 1982) and the real-valued Maxwell-Bloth equations (David and Holm, 1992). Under a certain range of initial conditions (ICs), all of the three systems can be reduced to become the simple ODE, $X''+X^3/2$, which is a special case of the Duffing equation. Therefore, we may conclude that the nonlinear restoring force that produces periodic solution also appears in the original 3D-NLM with a larger r and the real-valued Maxwell-Bloth equations.

Based on the npg-16-40 discussion article, we have extended our study to examine the impact of the extension of the nonlinear feedback loop on the recurrence of solutions using the 5D-NLM (Faghih-Naini and Shen, 2017, npg-17-2). The extended nonlinear feedback loop produces two incommensurate frequencies, leading to a quasi-periodic solution. The "recurrent" solution trajectory moves endlessly on a torus but never intersects itself. Additionally, Appendix C of Faghih-Naini and Shen (2017) provides an analogy between a coupled system with two springs and the locally linear 5D-NLM both of which have the same mathematical equation. This analogy helps reveal the role of the extended nonlinear feedback in producing quasi-periodic (recurrent) solutions and the importance of mode selection and coupling. Therefore, using the 3D-NLM and 5D-NLM, we have shown the role of the nonlinear feedback loop in producing periodic and quasi-periodic solutions, respectively. Since the nonlinear feedback loop and its extension appear throughout the spatial mode-mode interactions rooted in the nonlinear temperature advection, we expect that recurrent solutions such as quasi-periodic solutions may appear in the system based on the (full) Navier-Stokes equations. As "folding" is crucial for the occurrence of chaotic solution, future work will examine the relationship between the folding and recurrence in the high dimensional dissipative and non-dissipative Lorenz models.

In the npg-16-40 discussion article, we also illustrated how the collective impact of the nonlinear feedback loop and the heating may produce three types of solutions, including nonlinear periodic solutions with a small or large cycle and the homoclinic orbit solution. Additionally, we discussed the energy cycle. Note that kinetic, available potential energy (APE) and potential energy (PE), can be identified as $X^2$, $Y^2+Z^2$ and Z, respectively, in the 3D-NLM (e.g., Eqs. 10-12). However, simply considering

X"+$M^2$X=0 (Eq. 15) that is a the "special" Duffing equation, we can only define kinetic and potential energy, as follows: $KE_d = (1/2)\sigma^2 Y^2$ and $PE_d = (1/8)X^4 - (1/2)(\sigma r + C_1/C_0) X^2$, where $C_1/C_0$ is defined in Eq. 13. From a perspective of the 3D-NLM, "Z" is missing in the "special" Duffing equation. It is known that Z is crucial for introducing nonlinear terms in the 3DLM or 3D-NLM. Therefore, 3D-NLM can provide more detailed information about the energy transfer among the KE, PE and APE, as compared to the 2nd order ODE, X"+$M^2$X=0 or X"+$X^3$/2=0.

While specific responses are provided below, a summary on what has been discussed in re-revised manuscript is given as follows:

- Using a nondissipative Lorenz model, we present a closed-form solution using trigonometric functions to show that the nonlinear feedback loop (consisting of the nonlinear terms –XZ and XY) acts as a restoring forcing and the heating term alone can produce a saddle point.

- Using closed-form and numerical solutions, we showed that the nonlinear feedback loop and heating term collectively lead to three critical points and three types of solutions.

- Based on the energy analysis, a small energy cycle with four different regimes, which is half of a big energy cycle, is identified in one type of oscillatory solutions. We illustrated that the relative impact of the nonlinear restoring force and linear (heating) force determines the partitions of the averaged available potential energy associated with the Y and Z modes at different stages (i.e., linear and nonlinear stages). A big energy cycle appears in another type of oscillatory solutions. [Note that if we simply consider X"+$M^2$X=0 or X"+$X^3$/2=0, we can only define kinetic and potential energy.]

- The existence of the homoclinic orbit and two types of oscillatory solutions indicates the importance of the nonlinearity (i.e., the nonlinear feedback loop) and suggests the appearance of diverged trajectories. This type of solution dependence on ICs is different from the one associated with a chaotic attractor in the 3DLM.

- As suggested by one reviewer (for the npg-14-21 discussion article), Appendix A was added to discuss the derivations of Eq. (3) that includes the σY term;

- As suggested by one reviewer (for the npg-14-21 discussion article), Appendix C and Figure C1 were added to present the closed-form solution to the special Duffing

equation using elliptic functions and compare it with the closed-form solution represented by the elementary trigonometric functions in section 3.

- To respond to the comments on the npg-16-40 discussion article, Appendix B is added as a brief summary on what has been discussed in Shen (2014) to identify the nonlinear feedback loop. Additionally, we provided a comparison between the 3D-NLM, the 3DLM with a very large r and the real-valued Maxwell-Bloch equations.

**References:**

David, D. and D. D. Holm, 1992: Multiple Lie-Poisson Structures, Reductions, and Geometric Phases for the Maxwell-Bloch Travelling Wave Equations. J. Nonlinear Sci. 2, pp. 241-262.

Devaney, R. L. 1989: An Introduction to Chaotic Dynamical Systems. 2$^{nd}$ Edition. Addison Wesley, 336pp.

Faghih-Naini, S. and B.-W. Shen, 2017: On quasi-periodic solutions associated with the extended nonlinear feedback loop in the five-dimensional non-dissipative Lorenz model. Nonlin. Processes Geophys. Discuss. (npg-17-2, submitted in Jan., 2017).

Shen, B.-W., 2016: Hierarchical scale dependence associated with the extension of the nonlinear feedback loop in a seven-dimensional Lorenz model. Nonlin. Processes Geophys., 23, 189-203, doi:10.5194/npg-23-189-2016, 2016.

Shen, B.-W., 2015: Nonlinear Feedback in a Six-dimensional Lorenz Model. Impact of an additional heating term. Nonlin. Processes Geophys., 22, 749-764, doi:10.5194/npg-22-749-2015, 2015.

Shen, B.-W., 2014: Nonlinear Feedback in a Five-dimensional Lorenz Model. J. of Atmos. Sci., 71, 1701–1723. doi:http://dx.doi.org/10.1175/JAS-D-13-0223.1

Shen, B.-W. and S. Faghih-Naini, 2017: On recurrent solutions in high-dimensional non-dissipative Lorenz models. The 10th Chaos Modeling and Simulation International Conference (CHAOS2017), Barcelona, Spain, 30 May - 2 June, 2017. (Accepted for oral presentation, December 31, 2016).

Shen, B.-W., R. Atlas, O. Oreale, S.-J Lin, J.-D. Chern, J. Chang, C. Henze,and J.-L. Li, 2006: Hurricane Forecasts with a Global Mesoscale-Resolving Model: Preliminary Results with Hurricane Katrina(2005). Geophys. Res. Lett., L13813, doi:10.1029/2006GL026143.

Shen, B.-W. W.-K. Tao, Y.-L. Lin, A. Laing, 2012: Genesis of Twin Tropical Cyclones as Revealed by a Global Mesoscale Model: The Role of Mixed Rossby Gravity Waves. J. Geophys. Res., 117, D13114, doi:10.1029/2012JD01745.

Shen, B.-W., M. DeMaria, J.-L. F. Li, and S. Cheung, 2013: Genesis of Hurricane Sandy (2012) Simulated with a Global Mesoscale Model. Geophys. Res. Lett. 40. 2013, DOI: 10.1002/grl.50934.

Sparrow, C.: The Lorenz Equations: Bifurcations, Chaos, and Strange Attractors. Springer, New York. Appl. Math. Sci., 41, 1982.

> The paper concerns itself with a set of equations that have its origins on the standard quasigeostrophic equations for atmospheric flows. The analysis is competent but not novel, the outcomes are not remarkable. Unless the Author could suggest in very specific and complete terms what relevance these equations have to a physical problem of interest it would be difficult for this paper to have a readership.

As discussed in the general responses, our main goal is to illustrate the role of the nonlinear feedback loop (i.e., -XZ and XY) and its extension in producing recurrent solutions, including periodic solutions in the 3D-NLM and quasi-periodic solutions in the 5D-NLM. In the following, We provided brief discussions on the origins of various Lorenz models, including the Lorenz84 model that is more applicable to the quasi-geostrophic (QG) system.

The ODEs for the original Lorenz-63 model (i.e., 3DLM) were derived from the partial differential equations (PDEs) for Rayleigh Benard convection. The nondissipative version (i.e., 3D-NLM) was derived using the same PDEs but ignoring dissipative terms (as discussed in Appendix A of npg-16-40). The original PDEs are non-hydrostatic. The nonlinear advection of temperature in the PDEs produces the nonlinear terms (e.g., XY and –XZ) in the 3DLM. As the current trend is to develop global non-hydrostatic models at a resolution of 1km or finer, it is my belief that improving our understanding of the feedback by small processes within the original and high-dimensional Lorenz models can help understand the thermodynamic feedback associated the explicitly resolving convective processes in the next generation global models.

In 1980, using shallow water equations (with a constant of "f", the Coriolis parameter), Lorenz applied the Galerkin method to derive a model and transformed it into the 3D Lorenz model (i.e., the 3DLM) using a complicated transformation. Later, Lorenz published a paper with different ODEs for the QG system in 1984, which has been referred to as the Lorenz-84 model. However, to our best knowledge, we could not find any paper by Lorenz regarding related derivations. Here, we would like to provide the following notes for reviewers' information. In 2015, I personally checked with two

researchers (who jointly published a paper using the Lorenz-84 model) and learned that they could not find references by Lorenz about the derivations of the Lorenz84 model. By searching for literatures later, derivations of the Lorenz84 model can be found in Veen (2002), as shown in the extracted image below. Veen (2002) made the following comments (on page 11 of Veen, 2002), which are similar to what I learned:

"

*To the author's knowledge, no derivation of the Lorenz-84 model from atmospheric flow equations has been presented before. A rather ad hoc link was established by Wiin-Nielsen [1992, 1994], but in his work the reduction to three degrees of freedom is not based on physical or mathematical arguments. The link established here enables us to calculate the parameters in the Lorenz-84 model from the physical parameters in the filtered equations. As it turns out, one of the parameters comes out significantly different from its traditional, yet unmotivated, value. A continuation in this parameter relates the bifurcation diagram found at the traditional parameter value, presented in Shilnikov et al. [1995], to the one found at the physical value. The latter still bears resemblance to the bifurcation diagram of the six dimensional model, but the neutral saddle-focus transition is no longer there. Hence the route to chaos through a Shilnikov type bifurcation is absent. It is shown, that chaos through period doubling cascades, the Ruelle-Takens scenario and intermittency does occur in the Lorenz-84 model.*

"
* * *
**2. The Lorenz-84 general circulation model**

Like the Lorenz-63 model, the Lorenz-84 model is related to a Galerkin truncation of the Navier-Stokes equations. Where the '63 model describes convection, the '84 model gives the simplest approximation to the general atmospheric circulation at midlatitude. The approximation is applicable on an $f$-plane, placed over the North Atlantic ocean.

We can give a physical interpretation of the variables of the Lorenz-84 model: $x$ is the intensity of the westerly circulation, $y$ and $z$ are the sine and cosine components of a large traveling wave. The time derivatives are given by

$$\dot{x} = -y^2 - z^2 - ax + aF \tag{1.1}$$

$$\dot{y} = xy - bxz - y + G \tag{1.2}$$

$$\dot{z} = bxy + xz - z \tag{1.3}$$

where $F$ and $G$ are forcing terms due to the average north-south temperature contrast and the earth-sea temperature contrast, respectively. Conventionally we take $a = 1/4$ and $b = 4$.
* * *
*Figure: the extracted image shows the Lorenz-84 model (see Veen, 2002 for details).*

**References:**

Lorenz, E.N., 1980: Attractor sets and quasi-geostrophic equilibrium, *J. Atm. Sci.* Vol. 37, pp. 1685-1699.

Lorenz, E. N., 1984: "Irregularity: a fundamental property of the atmosphere," Tellus

36A , 98–110.

Veen, L. van, 2002: Time scale interaction in low-order climate models. Utrecht
   University Repository. (PhD Dissertation).
Veen, L.van, 2002:  "Baroclinic flow and the Lorenz-84 model," Int. J. Bifurcation
   Chaos, 13, 2117 (2003). DOI: http://dx.doi.org/10.1142/S0218127403007904

* Responses to the comments by reviewer II

> This manuscript discusses a model called by the author the non-dissipative Lorenz model. The model is related to the Lorenz 1963 model, but with several terms missing. It is apparently derived from the Boussinesq equations by a method similar to that of Lorenz. The model has a conserved quantity, and for a particular value of the conserved quantity, the model is equivalent to the Duffing equation with no damping and no forcing, i.e., frictionless motion in a double-well potential given by a symmetric 4th-degree polynomial. The analysis of periodic and homoclinic solutions of this model is not novel, and it is not clear to me that it offers significant insight into the physics of the Boussinesq equations.

Thanks for your comments. My responses are provided as follows:

(1) The ODEs were derived from the PDEs for Rayleigh-Benard convection. The Boussinesq approximation was used in the PDEs, which may be called the Boussinesq-type equations.

(2) Our goal is to reveal the role of nonlinear feedback loop in producing periodic solutions within the 3D-NLM (as well as quasi-periodic solutions using the 5D-NLM, e.g., Faghih-Naini and Shen, 2017). We discussed how the nonlinear feedback loop, consisting of -XZ and XY, may lead to the nonlinear restoring forcing term (i.e., $X^3$). Previously, we have shown that the nonlinear feedback loop is from the nonlinear advection of temperature and can be extended using additional high wavenumber modes.

(3) While it has been shown that the homoclinic orbit may appear under certain conditions within the 3DLM, we did not find any published papers discussing the homoclinic orbit using the non-dissipative Lorenz model (3D-NLM).

(4) We are aware that a further simplified 3D-NLM (e.g., with a particular set of initial conditions) and the Duffing equation (as discussed in the Appendix) (or double-well potential system) may be dynamically equivalent. However, while the former (the 3D-NLM) has three ODEs and the latter (with or without an external forcing) is a second-order ODE. [In general, a second-order ODE can be reduced to a set of two first-order ODEs.] Therefore, without providing a proof regarding a homeomorphism, we avoided a detailed comparison between the two systems. On the other hand, using the 3D-NLM (e.g., Eqs. 4-5), we were able to

discuss the energy cycle (e.g., the partitions of averaged available potential energy from Y and Z modes) and the challenge of performing the numerical integration for obtaining the homoclinic orbit solution (see also discussions in (5) below).

(5) As shown in Figure 7 for the solutions of homoclinic orbits as well as oscillatory solutions with different temporal scales, adaptive methods is effective to perform numerical integrations of a homoclinic orbit, implying that it may be challenging to determine the Lyapunov exponent numerically when an initial condition is on the homoclinic orbit.

(6) We have already extended the current study using the 3D-NLM to examine the impact of the extended nonlinear feedback loop on the quasi-periodicity of solutions in the 5D-NLM (Faghih-Naini and Shen, 2017, npg-17-2).

(7) Based on the above discussions, it is our belief that the original and high-dimensional non-dissipative Lorenz models are unique as compared to the double-well potential system or the Duffing equation. On the other hand, the similarity between the 3D-NLM and the Duffing equation can help verify the our results (including numerical solutions) and build our confidence in the validity of analytical and numerical methods for studying the role of the extended nonlinear feedback loop in producing quasi-periodic solutions in high-dimensional non-dissipative Lorenz models.

To be more specific, equation (15) in the manuscript is the Duffing equation discussed above; I find it peculiar that the manuscript does not mention Duffing until Appendix B. The Duffing equation has been studied extensively, and in the particular case of no damping and no forcing, its solutions are particularly well understood. The closed-form solution in the special case presented in Appendix B is also derived, for example, in the discussion below equation (23) of the article at: http://www.sciencedirect.com/science/article/pii/S0094114X14002079 Closed-form solutions for more general equations are derived in the articles at: http://www.sciencedirect.com/science/article/pii/S0307904X12002302 http://isidl.com/downloadfile/106810 The present manuscript focuses primarily on the analysis of the Duffing model, which I already well understand, and I don't feel that I learned anything from it about the Lorenz or Boussinesq equations.

Thanks for sharing the two references. We have cited the references in the main text of the revised manuscript. We wanted to point out that in addition to a citation to the Duffing oscillator equation in Appendix B/C, we also cited the paper by Roupas (2012) who provided more detailed discussions regarding the impact of parameters on solutions and their relationship with the Duffing oscillator equation. For example, we did state that two types of solutions associated various $C_1/C_o$ were discussed by Roupas (2012). [Here the constant $C_1$ represents the conservation of KE and PE, i.e., $C_1/C_o = X^2/2-Z$ (Eg. 13). The Duffing oscillator equation does not include a component which is similar to "Z" in the 3D-NLM.] Specifically, depending on the sign of ($\sigma r + C_1/C_o$), Eq. 15 is called the Duffing equation or two-well potential system.

For the suggested references, which have been cited in the revised manuscript, I would like to provide additional responses as follows:

1. In the newly added references (i.e., Elias-Zuniga, 2013; Starossek, 2015), we could not find discussions regarding the homoclinic orbit.

2. As compared to the analytical solutions in Elias-Zuniga (2013) and Starossek, (2015), our closed-form solutions in the main text have a simpler representation. It is our belief that the simplicity of our closed-form solutions (to the simple nonlinear ODE $X''+X^3/2$) using trigonometric functions can help illustrate the periodicity of the solutions. Additionally, discussions on the solution and Duffing equation in Appendix B/C using elliptic functions were used for verification. Numerical simulations using the 3D-NLM are computed and inter-compared with our closed-form solutions. The 3D-NLM can help discuss the partitions of

averaged potential energy from Y and Z modes. We are not sure if this kind of analysis for the energy cycle can be done using the Duffing equation.

3. The study in the npg-16-40 discussion article has been extended to reveal the role of the extended nonlinear feedback loop in producing quasi-periodic solutions within the 5D-NLM (Faghih-Naini and Shen 2017, npg-17-2) as well as the 7D-NLM (Shen and Faghih-Naini, 2017).

Finally, I would like to add the following to support our approach and conclusion. In the newly-added Appendix B, we listed a reference showing that the 3D-NLM with r=0 can represent the original 3DLM with a very large r. Additionally, we documented the similarity between the 3D-NLM with r=0 and the real-valued Maxwell-Bloch equations, which can be written as follows (Eq. 1.20 of David and Holm, 1992):

X'=Y,
Y'=XZ,
Z'=-XY.

The above can be reduced to become

$$X'' - CX + (1/2)X^3 = 0,$$

here C $(=(1/2) X^2 + Z)$ is a constant, representing a conservation law. Note that replacing (XZ) and (–XY) by (-XZ) and (XY), respectively, we can obtain a set of equations that are the same as the 3D-NLM with r=0. Therefore, based on our study, it is suggested that the pair of (XZ) and (−XY) serves as a restoring force in the Maxwell-Bloch equations.

**References:**

David, D. and D. D. Holm, 1992: Multiple Lie-Poisson Structures, Reductions, and Geometric Phases for the Maxwell-Bloch Travelling Wave Equations. *J. Nonlinear Sci.* Vol. 2: pp. 241-262 (1992).

---

## Author Comment (AC5) · 7 Mar 2017

**1   Introduction**

The following discussions derived from the conference article by Shen and Faghih-Naini (2017, accepted for oral presentation) are provided for review process. It is shown that the governing equations of the locally linear 7D-NLM are identical to those in the coupled system with three identical masses and three different springs.

**2   Seven-dimensional Non-dissipative Lorenz Model**

This section describes the governing equations for the seven-dimensional non-dissipative Lorenz model (7D-NLM) and the corresponding locally linear 7D-NLM. We will then compare the 7D-NLM and a coupled system with three identical masses and three different springs.

The 7D-NLM can be obtained by removing the dissipative terms of the 7D (dissipative) LM (7DLM; Shen, 2016), as follows:

$$\frac{dX}{d\tau} = -\sigma X + \sigma Y, \tag{1}$$

$$\frac{dY}{d\tau} = -XZ + rX - Y, \tag{2}$$

$$\frac{dZ}{d\tau} = XY - XY_1 - bZ, \tag{3}$$

$$\frac{dY_1}{d\tau} = XZ - 2XZ_1 - d_0 Y_1, \tag{4}$$

$$\frac{dZ_1}{d\tau} = 2XY_1 - 2XY_2 - 4bZ_1, \tag{5}$$

$$\frac{dY_2}{d\tau} = 2XZ_1 - 3XZ_2 - d_0 Y_2, \tag{6}$$

$$\frac{dZ_2}{d\tau} = 3XY_2 - 9bZ_2. \tag{7}$$

The same approach has been used to derive the 3D-NLM and 5D-NLM (e.g., Faghih-Naini and Shen, 2017). The dissipative terms are indicated by the terms with a crossout symbol. As discussed in Shen (2016), $(X, Y, Z, Y_1, Z_1, Y_2, Z_2)$ represent the amplitude of the Fourier modes. We refer to $(X, Y, Z)$ as the primary modes, $(Y_1, Z_1)$ as the secondary modes, and $(Y_2, Z_2)$ as the tertiary modes. $\tau$ is dimensionless time. The two parameters $(\sigma, r)$ are the Prandtl number and the normalized Rayleigh number (or the heating parameter), respectively. Detailed information regarding these parameters and ignored terms is provided in Shen (2016). On the right-hand side of the above equations, there are the linear heating term $(rX)$ and the nonlinear force terms (e.g., $-XZ$ and $XY$).

Applying a perturbation method, which represents the total field $(A)$ as a sum of the reference state $(A_c)$ and perturbation $(A')$, i.e., $A = A_c + A'$, we

transform Eqs. (1-7) to the following equations:

$$\frac{dX'}{d\tau} = \sigma Y', \tag{8}$$

$$\frac{dY'}{d\tau} = (r - Z_c)X' - X_c Z' - FN(X'Z'), \tag{9}$$

$$\frac{dZ'}{d\tau} = (Y_c - Y_{1c}) X' + X_c Y' - X_c Y_1' + FN(X'Y' - X'Y_1'), \tag{10}$$

$$\frac{dY_1'}{d\tau} = (Z_c - 2Z_{1c})X' + X_c Z' - 2X_c Z_1' + FN(X'Z' - 2X'Z_1'), \tag{11}$$

$$\frac{dZ_1'}{d\tau} = (2Y_{1c} - 2Y_{2c})X' + 2X_c Y_1' - 2X_c Y_2' + FN(2X'Y_1' - 2X'Y_2'). \tag{12}$$

$$\frac{dY_2'}{d\tau} = (2Z_{1c} - 3Z_{2c})X' + 2X_c Z_1' - 3X_c Z_2' + FN(2X'Z_1' - 3X'Z_2'), \tag{13}$$

$$\frac{dZ_2'}{d\tau} = 3Y_{2c}X' + 3X_c Y_2' + FN(3X'Y_2'). \tag{14}$$

As discussed in Shen (2014) and Faghih-Naini and Shen (2017), the flag $FN$ is introduced to perform linear simulations ($FN = 0$) or nonlinear simulations ($FN = 1$). Equations (1-7) are referred to the 7D-NLM V1 and Eqs. (8-14) are referred to the 7D-NLM V2. The 7D-NLM V1 and V2 should produce identical results with the same initial conditions except when round-off errors become different in the runs using different models. The V2 with FN=0 is also called the locally linear 7D-NLM, which can be used for the linear stability analysis.

**2.1  A comparison with the coupled system with three springs**

Choosing $FN = 0$ and $(Y_c, Z_c, Y_{1c}, Z_{1c}, Y_{2c}, Z_{2c}) = (0, r, 0, \frac{r}{2}, 0, \frac{r}{3})$, we can obtain:

$$\frac{d^2 Y'}{d\tau^2} = -X_c \frac{d^2 Z'}{d\tau^2} = -X_c^2 (Y' - Y_1') \tag{15}$$

from Eqs. (9-10),

$$\frac{d^2 Y_1'}{d\tau^2} = X_c \frac{d^2 Z'}{d\tau^2} - 2X_c \frac{d^2 Z_1'}{d\tau^2} = X_c^2 (Y' - 5Y_1' + 4Y_2') \tag{16}$$

from Eqs. (10-12),

$$\frac{d^2 Y_2'}{d\tau^2} = 2X_c \frac{d^2 Z_1'}{d\tau^2} - 3X_c \frac{d^2 Z_2'}{d\tau^2} = X_c^2 (4Y_1' - 13Y_2') \tag{17}$$

from Eqs. (12-14).

Previously, we have shown that the locally linear 3D-NLM and 5D-NLM have the governing equations that are identical to the systems with one spring and two springs, as shown in Figures 1a and 1b, respectively. For a comparison with the 7D-NLM, we present the governing equations for the coupled system with three identical masses and three different springs, as shown in Figure 1c.

$$\frac{d^2 x_1}{d\tau^2} = -k_1(x_1 - x_2) \tag{18}$$

$$\frac{d^2 x_2}{d\tau^2} = -k_2(x_2 - x_3) - k_1(x_2 - x_1) \tag{19}$$

$$\frac{d^2 x_3}{d\tau^2} = -k_3 x_3 - k_2(x_3 - x_2) \tag{20}$$

The top, middle, and bottom springs have spring constants of $k_3$, $k_2$, $k_1$, respectively. Here, the top spring is attached to the ceiling on one end and to the top mass on the other end. The upper (low) end of the middle spring is attached to the top (bottom) mass. For the bottom spring, its upper end is attached to the middle mass. $x_1(\tau)$, $x_2(\tau)$ and $x_3(\tau)$ are the displacements of the centers of masses from equilibrium. By choosing $x_1 = Y'$, $k_1 = X_c^2$, $x_2 = Y_1'$, $k_2 = 4X_c^2$, $x_3 = Y_2'$ and $k_3 = 9X_c^2$, we show that Eqs. (18-20) are identical to Eqs. (15-17), respectively. In other words, the above coupled system with three springs is identical to the locally linear 7D-NLM. Note that for each of uncoupled one-mass-one-spring systems, the frequency of the oscillatory motion is either $X_c$, $2X_c$, or $3X_c$. By comparisons, in section 3.1, we will show that the above system have three frequencies, but they are different from the values of $X_c$, $2X_c$, or $3X_c$. More importantly, these frequencies are incommensurate, leading a quasi-periodic solution. As the 7DLM (or 7D-NLM) is derived by properly selecting new modes to extend the nonlinear feedback loop of the 5DLM (or 5D-NLM), we will discuss how the extended nonlinear feedback loop introduces two additional pair of downscaling and upscaling processes to produce an additional temporal oscillatory mode that is coupled with existing two temporal oscillatory modes.

**References**

1. Faghih-Naini, S. and B.-W. Shen, 2017: On quasi-periodic solutions associated with the extended nonlinear feedback loop in the five-dimensional non-dissipative Lorenz model. Nonlin. Processes Geophys. Discuss. (submitted Jan. 16, 2017).
2. Shen, B.-W. and S. Faghih-Naini, 2017: On recurrent solutions in high-dimensional non-dissipative Lorenz models. The 10th Chaos Modeling and Simulation International Conference (CHAOS2017), Barcelona, Spain, 30 May - 2 June, 2017. (accepted, December 30, 2016)

[Figure]

Figure 1: Systems with one mass and one spring (a), two masses and two springs (b) and three masses and three springs (c). Three masses are identical, i.e., $m_1 = m_2 = m_3$. Three spring constants $k_1$, $k_2$ and $k_3$ are selected as $X_c^2$, $4X_c^2$, and $9X_c^2$, respectively. It is shown that the governing equations for the above systems in panels (a)-(c) are identical to those for the locally linear 3D-NLM, 5D-NLM, and 7D-NLM, respectively. This comparison illustrates how the nonlinear feedback loop and its extension enabled by a proper selection of high wavenumber modes can produce recurrent (i.e., periodic or quasi-periodic) solutions.